# A Comparative Event-Related Potentials Study between Alcohol Use Disorder, Gambling Disorder and Healthy Control Subjects through a Contextual Go/NoGo Task

**DOI:** 10.3390/biology12050643

**Published:** 2023-04-24

**Authors:** Macha Dubuson, Xavier Noël, Charles Kornreich, Catherine Hanak, Mélanie Saeremans, Salvatore Campanella

**Affiliations:** 1Laboratoire de Psychologie Médicale et d’Addictologie, Faculty of Medicine, Université Libre de Bruxelles (ULB), CHU Brugmann, Psychiatry Institute, 4 Place Vangehuchten, ULB Neuroscience Institute (UNI), 1020 Brussels, Belgium; macha.dubuson@ulb.be (M.D.); xavier.noel@ulb.be (X.N.); charles.kornreich@chu-brugmann.be (C.K.); catherine.hanak@chu-brugmann.be (C.H.); melanie.saeremans@chu-brugmann.be (M.S.); 2Haute Ecole Provinciale de Namur, 5000 Namur, Belgium

**Keywords:** addiction, event-related potential, inhibition response, Go/NoGo task, alcohol use disorder, gambling disorder

## Abstract

**Simple Summary:**

Many societies report a high number of people suffering from behavioral or substance-related addictions, such as gambling or alcohol. Despite psychotherapy, social support, withdrawal, or even medication, it is recognized throughout the world that recovering from an addiction is particularly challenging. Understanding the neurocognitive mechanisms triggering addictive disorders is therefore particularly relevant to optimizing addiction treatment. In the present study, we investigated whether or not patients suffering from gambling or alcohol use disorders are efficient at inhibiting their responses when their attention is attracted by a neutral, rewarding, or cueing context related to their own addiction (alcohol vs. gambling). Such behavioral and neural evidence may help clinicians to implement novel targeted intervention more suited to the individual needs of these patients.

**Abstract:**

(1) Background: Inhibitory and rewarding processes that mediate attentional biases to addiction-related cues may slightly differ between patients suffering from alcohol use (AUD) or gambling (GD) disorder. (2) Methods: 23 AUD inpatients, 19 GD patients, and 22 healthy controls performed four separate Go/NoGo tasks, in, respectively, an alcohol, gambling, food, and neutral long-lasting cueing context during the recording of event-related potentials (ERPs). (3) Results: AUD patients showed a poorer inhibitory performance than controls (slower response latencies, lower N2d, and delayed P3d components). In addition, AUD patients showed a preserved inhibitory performance in the alcohol-related context (but a more disrupted one in the food-related context), while GD patients showed a specific inhibitory deficit in the game-related context, both indexed by N2d amplitude modulations. (4) Conclusions: Despite sharing common addiction-related mechanisms, AUD and GD patients showed different patterns of response to (non-)rewarding cues that should be taken into account in the therapeutic context.

## 1. Introduction

The ability to stop the execution of a spontaneous or planned reaction is called “response inhibition” [1]. This executive control mechanism is notably a core process allowing people to inhibit impulsive responses to stimuli. When impaired, it can trigger impulsive responses which are well-known to characterize behavioral (e.g., gambling disorder, GD) [2] as well as substance-related (e.g., alcohol use disorder, AUD) [3] addictive disorders. This altered inhibitory skill and an increased salience of addiction-related cues are the two main disturbed mechanisms diverting the patients’ attention and triggering an overtrained pattern of substance use [4,5]. Indeed, addictive stimuli were associated through classical conditioning stimuli to rewarding and highly motivational responses (such as excessive alcohol consumption or gambling) [6]. Such stimuli will therefore attract patients’ attention and elicit conditioned responses [7] (a consumption behavior that has become dominant over time and that cannot be regulated or stopped). In other words, according to the incentive-sensitization theory (IST) [8], the repeated exposures to addiction-related stimuli (gambling or alcohol) sensitize the dopaminergic response in brain reward areas, enhancing the incentive-motivational properties of these cues through associative learning. These salient stimuli attract consumers’ attention (generating an attentional bias), acquire a highly motivational value, and guide behavior toward consumption. Such a view has been conceptualized by Goldstein and Volkow [9,10] in the Impaired Response Inhibition and Salience Attribution (I-RISA) model, a model substantiated by many empirical behavioral studies, disclosing by means of numerous brain imaging studies specific impairments in six large-scale brain networks [11], and supporting behavioral as well as substance-related addictions [12]. As an add-on tool to psychotherapy, social support, and medication, such findings led to the consideration of these mechanisms as key targets to be rehabilitated in GD and AUD patients in order to promote abstinence and well-being [13].

If the unbalance between (hypoactive) inhibitory and (hyperactive) attentional mechanisms clearly impacts both GD and AUD, several studies have been devoted to verifying whether response inhibition in GD and AUD patients is exactly altered in a similar way [14]. Indeed, because of alcohol neurotoxicity [15], the brain regions involved in response inhibition could be more impaired in AUD than in GD patients. Laboratory studies have shown that, compared with healthy controls, both GD and AUD patients showed a worse inhibitory performance [16], even if AUD patients appeared to be even more impaired (slower reaction times, attenuated post-error slowing), probably due to the deleterious effect of alcohol on the fronto-striatal circuitry [17]. However, the way attentional resources are impacted in GD and AUD could also be questioned. Indeed, these attentional biases can be interpreted either as depending on “a global blunted reward system”, in which addicted patients seek intense rewarding behaviors as a compensation mechanism [18], or as “a specific motivational bias”, meaning that GD and AUD patients attribute a more intense response to their respective addiction-related reward (gambling vs. alcohol stimuli) [19]. In real life, the ability to learn appropriate stimulus–response–outcome associations is of the greatest importance to identifying contingencies between specific behaviors and rewarding or damaging outcomes [20]. This contingency mechanism seems clearly impaired both in GD and AUD patients. Indeed, for instance, while gamblers showed a differential sensitivity to monetary versus non-monetary rewards at both the motivational and hedonic levels [21], alcohol drinkers showed a robust incentive-motivational value of alcohol compared with naturally rewarding activities (erotica, adventure scenes) [22]. This dysfunctional reward valuation has been clearly associated with an increased risk of developing AUD as well as GD, even if differential effects have been observed, particularly for loss avoidance, between these two patient populations [23].

Overall, several studies have strengthened the idea that, even if the inhibitory and rewarding processes that mediate attentional biases to addiction-related cues may slightly differ between GD and AUD patients, dysfunctional global similarities between these two groups also suggest a common neurocognitive aetiology for these disorders [16]. A classical way to investigate the interaction between these neurocognitive mechanisms has focused on the recording of event-related potentials (ERPs) during contextual Go/NoGo tasks [24,25,26,27,28,29,30]. As a result of their optimal temporal resolution, ERPs offer the possibility to capture the different stages involved in the information processing stream of a task, and therefore to infer the impaired neurocognitive stage when a performance is altered [31]. Applied to a covert process (e.g., producing no overt measurable behavior when successful) such as inhibition, ERPs measured in Go/NoGo tasks revealed, over fronto-central sites, a negative (N2) and a positive potential (P3) in NoGo trials as compared with Go ones in an interval ranging from 250 to 500 ms. At a functional level, these components are thought to reflect conflict monitoring and motor inhibition, respectively. Importantly, the “difference” wave (NoGo minus Go trials), indexed the “inhibitory” Go/NoGo effect per se, indexed by NoGo N2d and NoGo P3d components. Because context has been shown to modulate these inhibitory processes [32], contextual Go/NoGo tasks have been introduced, in which Go and NoGo trials were presented on long-lasting cueing backgrounds. Such backgrounds could therefore be adapted to specific addictions, such as alcohol [30,33], nicotine [26], gambling [24,25], or polydrug consumption [28]. This way, the impact of the attentional resources devoted to the cueing background on the inhibitory process could be measured. For instance, heavy social drinkers and gamblers were shown to make more inhibitory errors than light drinkers or controls, but only in the appropriate (alcohol/gambling)-related context, while at the neurophysiological level, this was reflected by a delayed NoGo P3 component [30] or a decreased NoGo N2 [25]. In the present study, and for the first time to our knowledge, AUD and GD patients will be confronted during an EEG recording to a set of four contextual Go/NoGo tasks, including a neutral, an alcohol-related, a gambling-related and another rewarding (food) cueing background, and compared with healthy controls. The main behavioral hypothesis is that AUD and GD patients will make more commission errors than healthy controls, with a higher amount/number of errors in their specific addiction-related context. ERPs will allow us to monitor the neurophysiological origin of this behavior along the information processing stream, to index potential differences between AUD and GD patients, and to observe the impact of another naturally rewarding context (such as food) on the inhibitory process of these two populations.

## 2. Materials and Methods

### 2.1. Participants and Ethics Statement

A total of 86 subjects took part in this study (Figure 1). In total, 64 subjects were included in the final analyses, 18 women and 46 men, aged between 24 and 65 years (mean = 40 years, SD = 10.53, 95%). Recruitment for the study took place between 2016 and 2020. The remaining 22 subjects were excluded since their behavioral data deviated by more than two standard deviations from the mean and/or for poor EEG signal-to-noise ratio.

Patients were recruited at the Brugmann University Hospital in Brussels, Belgium. The hospital’s ethics committee approved our study (CE 2016/121). Inpatients suffering from AUD (*n* = 23, 8 women/15 men, age mean = 47.52, SD = 8.37) were included during a 4-week alcohol rehabilitation. During the hospitalization, the inpatients received bio-psycho-social support and medication including a decreasing dose of diazepam (Day 1–5: 50 mg per day, then a decrease of 10 mg every two days) and one dose per day of Befact vitamins (B1-B2-B6-B12). The experiment took place during the end of the second week (Day 11), meaning that patients were taking under 20 mg of diazepam on average. Outpatients suffering from GD (*n* = 19, 1 woman/18 men, age mean = 39.68, SD = 8.79) who participated in our study were under psychological follow-up at the Gambling Clinic of CHU Brugmann.

The inclusion criteria targeted French speakers between 18 and 65 years old with severe AUD requiring alcohol rehabilitation or severe GD with psychological treatment, and a desire to reduce their use. Affective disorders were allowed for AUD and GD patients. The exclusion criteria were a personal history of a neurological disorder, diagnosis of a chronic psychotic disorder, use of alcohol or other illicit substance during the experiment, and hair incompatibility with EEG.

Healthy control subjects (*n* = 22, 9 women/13 men, age mean = 32.45, SD = 8.46) were enrolled as paid volunteers (20 €). Exclusion criteria for healthy control subjects were having a personal neurological or psychiatric history and hair incompatibility with EEG.

### 2.2. Procedure

The experiment was proposed to AUD patients at the end the end of the second week of their hospitalization, when they were receiving a minimal dose of diazepam. It was offered to GD patients by their psychologist during their follow-up at the clinic. Healthy control subjects were recruited through social media. All subjects were informed about the study and signed consent forms. They were free to leave at any time.

Participants were asked to answer some sociodemographic questions, to fill out the questionnaires, and to take part in the Mini International Neuropsychiatric Interview (MINI) [34].

The battery of questionnaires was as follows:The *Beck Depression Inventory* (BDI-II) [35] to score the severity of depression symptoms (21 items; range, 0–63). A score of 10–18 indicates mild depression; a score of 19–29 indicates moderate depression; and a score of 30–63 indicates severe depression.The *Alcohol Use Disorder Identification Test* (AUDIT) [36] to score the severity of alcohol use (10 items; range, 0–40). Problematic alcohol use is assumed for a score higher than 12 for males and 11 for females.The *South Oaks Gambling Screen* (SOGS) [37] to score the severity of gambling use (20 items; range, 0–20). Problematic gambling practice is considered for a score of 5 or more.The *Craving Experience Questionnaire* (CEQ) [38,39] to score the intensity and frequency of craving in the previous week (22 items; range, 22–154). The version of the CEQ was chosen based on the patient’s addiction to alcohol or gambling.

### 2.3. Contextual Go/NoGo Tasks

While sitting in front of a laptop (14-inch screen), participants were asked to stare at the screen, relax, and avoid moving as much as possible. The instruction was to press a button on a joystick with the thumb of the right hand, as quickly and accurately as possible, when the letter M (Go) was displayed, and not to press the button when the letter W (NoGo) was displayed.

The task consisted of four Go/NoGo blocks of 200 stimuli each (140 Go and 70 NoGo). Each block had a different context generated by an image of alcohol, gambling, food, or a neutral context displayed in the background throughout each block (Figure 2). The order of the contexts was counterbalanced across participants. The Go or NoGo (M or W) signal was presented for 200 ms, and then the signal disappeared for 1300 ms, leaving 1300 ms for the participant to respond. The order of stimulus presentation was pseudorandomized, so that no two NoGo were presented after each other and no more than four Go were presented before a NoGo.

Three categories of behavioral data are recorded: correct detection (successful Go response, i.e., when the participant presses the button after a Go signal); reaction time for the Go response, and commission error (unsuccessful NoGo response, i.e., when the participant presses the button after a NoGo signal).

### 2.4. EEG Recording and Treatment

During the contextual Go/NoGo tasks, the electroencephalograms (EEG) activity was recorded with 32 Quick-Cap electrodes (32-channel EEG cap waveguard™connect, ANT Neuro, Eenschede, The Netherlands), according to the 10–20 system and intermediate positions (Fpz, Fp1, Fp2, Fz, F3, F7, F4, F8, FC1, FC5, FC2, FC6, Cz, C3, C4, T7, CP5, CP1, CP2, CP6, T8, P7, P3, Pz, P4, P8, POz, O1, Oz, and O2), and with a linked mastoid physical reference (M1, M2). The signal was augmented by battery-operated ANT^®^ amplifiers with a gain of 30,000 and using a bandpass filter of 0.01–100 Hz. The electrode impedance was maintained below 10 kohm. The EEG was recorded continuously at a sampling rate of 1024 Hz with ANT^®^ EEprobe software.

Treatment of EEG activity consisted of applying a 0.3–30 Hz bandpass filter and creating 1000-ms stimulus-locked epochs, precisely 200 ms before and 800 ms after the signal (baseline from −200 ms to 0). Trials contaminated by eye movements or muscle artifacts were automatically eliminated offline (cutoff of 30 mV). The trials were averaged with a minimum of 15 trials for both successful Go and NoGo trials. Once averages were computed, the related peak amplitude and latency of the “Go” and “NoGo” ERP components N2 NoGo, P3 NoGo, N2 Go, and P3 Go were observable in each participant and for each context. The N2 component is identified as the largest negative peak that appears between 200 and 400 ms after stimulus display and the P3 component is identified as the largest positive peak between 300 and 600 ms after stimulus display. The component values were measured and represent the average of frontocentral electrodes (Fz, FC1, FC2, Cz). In addition, we calculated the N2d and P3d by the subtraction of the averaged “NoGo minus Go”, respectively, for the N2 and P3 components. As no “group” statistical differences emerged for the Go stimuli, and because NoGo N2d and P3d reflected inhibitory processes per se [28], all further analyses were conducted on the NoGo N2d and P3d components.

### 2.5. Statistics

All statistical analyses were conducted using IBM SPSS Statistics (v. 26). The threshold for significant effects was *p* < 0.050. Demographic and clinical variables were compared between the three groups, by using one-way analysis of variance (ANOVA) and Pearson chi-square (Table 1). Behavioral variables, such as correct detection rates, the reaction times, and the commission error rates, gathered during the four contextual Go/NoGo tasks, were analyzed using mixed ANOVAs (Groups [AUD, GD, controls] × Contexts [alcohol, game, food, neutral]). Neurophysiological variables (latency and amplitude of N2d and P3d) in the contextual Go/NoGo were analyzed using mixed ANOVAs (Groups [AUD, GD, controls] × Contexts [alcohol, game, food, neutral]). All mixed ANOVAs were performed using Greenhouse–Geiser corrections when applicable. Age, gender, and education variables were included as covariates. In order to understand significant effects, independent samples *t*-tests and paired samples *t*-tests were performed. In addition, Pearson correlations were performed on the behavioral and neurophysiological data.

## 3. Results

### 3.1. Clinical Data

One-way ANOVA showed a significant difference of age between the three groups (*F* = 17.852, *p* = < 0.001). Post hoc Bonferroni *t*-tests showed that AUD patients were older than GD patients (*p* = 0.013), AUD patients were older than controls (*p* < 0.001) and GD patients were older than controls (*p* = 0.026).

One-way ANOVA showed a significant difference of number of years of education (*F* = 17.852, *p* = < 0.001). Post hoc Bonferroni *t*-tests showed that GD patients had fewer years of education than controls (*p* = 0.031) and no significant difference was found when comparing with the other groups.

Chi-square analyses showed a significant difference of gender (*F* = 7.195, *p* = 0.027). Additional chi-square analyses showed that there were fewer females among the GD patients than among the AUD patients (*p* = 0.020) and controls (*p* = 0.008). No significant difference was found between AUD patients and controls (*p* = 0.672).

These three variables (age, gender, and education) are important differences between groups, as they could influence inhibitory ERP components [40,41,42]. We therefore included them as covariates in the following analyses.

### 3.2. Reaction Time Results

The results of mixed ANOVAs with age, sex, and education level as covariates conducted on the reaction time showed a significant main effect of the group (*F*(2, 58) = 5.050, *p* < 0.010, ηp^2^ = 0.148). Independent samples *t*-tests revealed that AUD patients had a slower reaction time than controls (*t*(*43*) = 3.227, *p* = 0.002) and GD patients (*t*(*40*) = 4.259, *p* < 0.001). There was no significant difference between GD patients and controls (*t*(*39*) = 0.227, *p* = 0.227).

Despite a lower reaction time in correct detection in the behavioral data for AUD patients compared with GD patients and controls, mixed ANOVA with covariates (sex, age and education level) conducted on the Go P3 latency showed no significant effect of group (*F*(2, 58) = 0.273, *p* = 0.762, np^2^ = 0.009).

Regarding the neurophysiologic components of inhibition, mixed ANOVA with covariates (sex, age, and education level) conducted on the P3d latency revealed a main effect of group (*F*(2, 58) = 3.394, *p* = 0.040, np^2^ = 0.105).

Independent samples *t*-tests (illustrated in Figure 3 and Figure 4) conducted on the P3d latency mean of the four contexts (AUD patients mean = 429.52, SD = 37.53; GD patients mean = 400.79, SD = 37.55; controls mean = 407.38, SD = 26.53) revealed a later latency for AUD patients than for GD patients (*t*(*40*) = 2.469, *p* = 0.018) and for controls (*t*(*43*) = 2.276, *p* = 0.028), although there was no significant difference between GD patients and controls (*t*(*39*) = −0.656, *p* = 0.516).

### 3.3. Performance Results

The results of mixed ANOVAs with age, sex, and education level as covariates conducted on the commission error rates showed a significant effect of the interaction group × context (*F*(6, 174) = 2.163, *p* = 0.049, ηp^2^ = 0.069). There was no significant effect of the context (*F*(3, 174) = 0.006, *p* = 0.94), the group (*F*(2, 58) = 1.589, *p* = 0.21), or the covariates (age: *F*(1, 58) = 0.001, *p* = 0.982; gender: *F*(1, 58) = 0.380, *p* = 0.54; education level: *F*(1, 58) = 0.547, *p* = 0.46).

Independent samples *t*-tests showed no significant difference between AUD and GD patients (*p* > 0.05; ddl = 40). Paired samples *t*-tests (illustrated in Figure 3) showed that controls did not make significantly more commission errors in any context (*p* > 0.05). Two main behavioral effects highlighted the group x context interaction described above and will be presented below: (i) compared with controls, an unexpected, enhanced number of commission errors occurred in the food context for AUD patients, and (ii) GD patients made more errors in the game context (as expected). At the neurophysiological level, significant results are obtained only through a mixed ANOVA with covariates (sex, age, and education level) conducted on the N2d amplitude, which showed a marginal main effect of group (*F*(2, 58) = 3.09, *p* = 0.053, np^2^ = 0.096). Independent samples *t*-tests conducted on the N2d amplitude mean of the four contexts (AUD patients mean = −3.03, SD = 2.43; GD patients mean = −3.04, SD = 2.16; controls mean = −4.61, SD = 2.40) revealed a smaller N2d amplitude for AUD patients (*t*(*43*) = 2.197, *p* = 0.033) and for GD patients (*t*(*39*) = 2.185, *p* = 0.035) than for controls. There was no significant difference between AUD patients and GD patients (*t*(*40*) = 0.919, *p* = 0.985). Data concerning N2d latencies or P3d were not statistically significant (see Figure 3 for illustration).

#### 3.3.1. AUD Patients Exhibit a Higher Commission Error Rate in the Food Context but Not in the Alcohol Context

Paired samples *t*-tests (illustrated in Figure 3) showed that AUD patients made more commission errors in the food context than in other contexts (compared with alcohol context *t* = −3.006, *p* = 0.007; game context *t* = 2.666, *p* = 0.014; neutral context *t* = −2.871, *p* = 0.009). At the neurophysiological level, paired samples *t*-tests revealed that AUD patients had higher N2d amplitude in the food context than in the neutral context (*t* = −2.318, *p* = 0.030).

Independent samples *t*-tests revealed higher commission error rates for AUD patients than controls in the food context (*t*(*43*) = 2.666, *p* = 0.011), the game context (*t*(*43*) = 2.041, *p* = 0.047), and the neutral context (*t*(*43*) = 2.023, *p* = 0.049). There was no significant difference in the alcohol context (*t*(*43*) = *0.905*, *p* = *0.370*). Regarding N2d component, independent samples *t*-tests revealed that AUD patients had significantly smaller N2d amplitude than controls in the game context (*t*(*43*) = 2.053, *p* = 0.046) and the neutral context (*t*(*43*) = 2.116, *p* = 0.040).

#### 3.3.2. GD Patients Exhibit a Higher Commission Error Rate in the Game Context

Paired samples *t*-tests (illustrated in Figure 3) showed that GD patients made more commission errors in the game context than in the neutral context (*t* = 2.985, *p* = 0.008). At the neurophysiological level, the paired samples *t*-test revealed that GD patients had smaller N2d amplitude in the game context than in the neutral context (*t* = 2.570, *p* = 0.019). The N2d amplitude was also smaller for the alcohol context than with the neutral context (*t* = 3.061, *p* = 0.007), and there was no significant difference between the food and neutral contexts (*t* = 1.224, *p* = 0.237).

Independent samples *t*-tests revealed higher commission error rates for GD patients than controls in the game context (*t*(*39*) = 2.444, *p* = 0.019), but not in the other contexts (neutral context *t* = 1.290, *p* = 0.205; alcohol context *t* = 1.291, *p* = 0.204; food context *t* = 1.261, *p* = 0.215). Independent samples *t*-tests on the N2d component revealed that GD patients had significantly smaller N2d amplitude than controls in the game context (*t*(*39*) = 2.531, *p* = 0.016) and the alcohol context (*t*(*39*) = 2.416, *p* = 0.020).

### 3.4. Correlations

To facilitate the interpretation of the ERP data, we performed Pearson correlation analyses between the behavioral and ERP data across all 64 participants, including the three groups. The results demonstrated a significant positive correlation between reaction time and P3d latency (*r* = 0.312, *p* = 0.012), indicating that the longer reaction times were linked with delayed P3d latency. Moreover, a negative correlation was found between the reaction time and commission error rate (*r* = −0.339, *p* = 0.006), suggesting that participants may have slowed down their responses to avoid making more errors.

## 4. Discussion

The main aim of the present ERP study was, for the first time to our knowledge, to compare the neurophysiological correlates of inhibitory processes when AUD and GD patients are performing a Go/NoGo task including a long-lasting context triggering a neutral, natural rewarding (food), alcohol-related, or gambling-related cued context. Three main findings were reported and are successively discussed below.

First, AUD patients were found to have slower reaction times than GD patients and controls in response to the Go trials, indexed by a slower P3d latency. The slower the reaction time, the more the P3d latency will be delayed. This suggests that participants use slowing reaction times as a compensatory mechanism, allowing AUD patients to reduce inhibitory errors compared with controls and GD patients [43]. As such, and as showed in previous studies, this suggests that AUD patients showed a less efficient inhibitory function than healthy people [33], indexed at the ERP level by a reduced cognitive inhibitory (N2d) as well as a delayed motor inhibitory (P3d) process. Such an effect could probably be accentuated by the use of diazepam during withdrawal of AUD patients [44].

Second, compared with controls, AUD patients had a higher rate of commission errors in the food context than in both the neutral and game contexts, whereas no difference between the groups was found for the alcohol context. At the neurophysiological level, this higher rate of commission errors in AUD patients was indexed by a generally decreased N2d amplitude compared with controls, with a higher N2d amplitude observable among AUD patients when they were confronted with the food context compared with the neutral context. Such data have two main implications. First, cognitive dual models predicted that AUD patients would show more commission errors and reduced NoGo ERP components when performing inhibition in an alcohol-related context [30,45]. This absence of effect was already reported in some previous studies [33,46], suggesting that alcohol-cue reactivity might be reduced or even inverted through detoxification. This could be neurophysiologically indexed by a recovery of N2d neural resources (cognitive inhibition), as AUD patients did show similar N2d amplitude to controls in the alcohol context. Second, the food context seems to hold a specific status among AUD patients as it generated more commission errors compared with controls, as well as a preserved N2d component (compared with the neutral and game contexts). These results are in line with recent data showing that alcohol can increase a food rewarding value and modulate a food-related attentional bias [47]. This relationship between food- and alcohol-related attentional biases deserves to be further investigated, as addictions share common rewarding pathways, and could therefore lead abstinent AUD patients to be (or not?) more vulnerable to developing a food addiction [48].

Third, compared with controls, GD patients had a higher rate of commission errors only in the game context. In addition, GD patients also had a higher rate of commission errors in the game context than in the neutral context. This was indexed by a specific decrease of N2d amplitude for stimuli presented in the game context, reflecting lower recruitment of neural resources involved in the detection and management of incongruence in contextual Go/NoGo tasks. Such data are perfectly matched with the dual theory idea and existing previous data [25,27,49] suggesting that the inhibitory skills of GD patients specifically decreased in a gaming context.

Overall, these data again highlight the importance of comparing different clinical populations with controls in the same experiment. By using a Go/NoGo task and diverse cueing contexts, our data suggest that, while AUD patients show a general impairment of inhibition irrespective of the neutral vs. rewarding nature of the used stimuli, GD patients seem to suffer from a more circumscribed alteration for gambling-related cues. Such data have a clear impact at the clinical level. These results suggest that treatment interventions should be tailored to the specific addiction-related cues that are most problematic for each patient. For example, GD patients may benefit from interventions that target their ability to inhibit responses to gambling-related cues, while AUD patients may benefit from interventions that help them to maintain inhibitory control in the presence of food-related cues. By recognizing and addressing these differences, treatment providers can help to optimize outcomes for patients with addiction-related disorders. Indeed, clear overlaps exist between behavioral and substance-related addictions in phenomenology, epidemiology, comorbidity, neurobiological mechanisms, genetic contributions, responses to treatments, and prevention efforts, but differences also exist [50]. A study of one million subjects found genes common to addiction and genes specific to different types of addiction, which may support the neurophysiological signature hypothesis [51]. Recognizing such criteria is therefore important in order to increase awareness of these disorders and to further develop prevention and treatment strategies.

There are several limitations that should be taken into account. One major limitation is the small sample size, which may limit the generalizability of the findings. In addition, the study included patients involved in a healthcare process and with some exclusion criteria. This may limit the generalizability of the findings to other populations suffering from addiction disorders, for example patients with comorbidities. Further studies should be undertaken in larger samples to confirm and extend these results. It may be interesting to observe patients suffering from AUD and GD at the same time. Another limitation is the limited scope of the study, which examined only specific types of cues (alcohol, gambling, food, and neutral cues) using a Go/NoGo task and ERPs. This may not fully capture the complexity of addiction-related processes and may overlook other factors that contribute to addiction. For a better understanding of addiction, larger studies with more variables would be needed. Finally, the study provided only cross-sectional data and did not examine changes in inhibitory and rewarding processes over time, which could provide a more comprehensive understanding of addiction-related processes and their course of development.

## 5. Conclusions

The objective of this study was to investigate how patients with AUD or GD are able to inhibit their responses to addiction-related cues in different contexts. By examining the neurocognitive mechanisms underlying addictive behaviors in these patient populations, we aimed to provide new insights that could help to optimize addiction treatment and develop more targeted interventions for individual patient needs. Our findings suggest that AUD and GD patients may have different patterns of response to (non-)rewarding cues, which highlights the importance of tailoring addiction treatment to the specific needs of each patient.

Based on the results of this study, it appears that patients suffering from AUD and GD have poorer inhibitory performance than healthy controls. However, there were some differences between the two patient groups in terms of their response to addiction-related cues. AUD patients showed a preserved inhibitory performance in the alcohol-related context, while GD patients showed a specific inhibitory deficit in the game-related context.

These findings suggest that inhibitory and rewarding processes may mediate attentional biases to addiction-related cues differently between patients with AUD and GD. Understanding these differences could help clinicians to develop more targeted interventions for addiction treatment that take into account individual patient needs. By identifying the neurocognitive mechanisms underlying addictive behaviors, these findings could help to optimize addiction treatment and improve outcomes for patients with these disorders.

## Figures and Tables

**Figure 1 biology-12-00643-f001:**
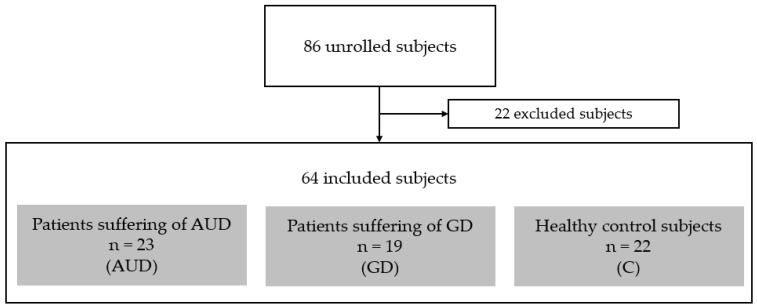
Enrolled subjects in the study. Abbreviations: AUD, alcohol use disorder; GD, gambling disorder; C, healthy controls.

**Figure 2 biology-12-00643-f002:**
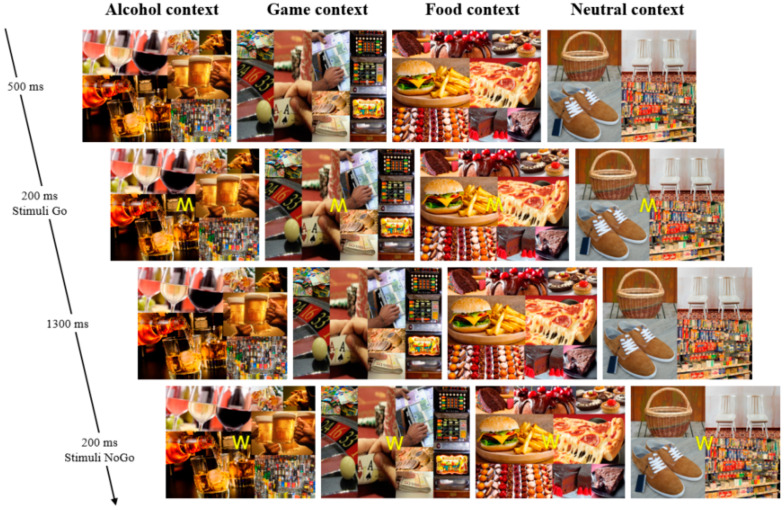
Illustration of the four contextual Go/NoGo tasks (alcohol, game, food, and neutral contexts), with Go (M) and NoGo (W) signals.

**Figure 3 biology-12-00643-f003:**
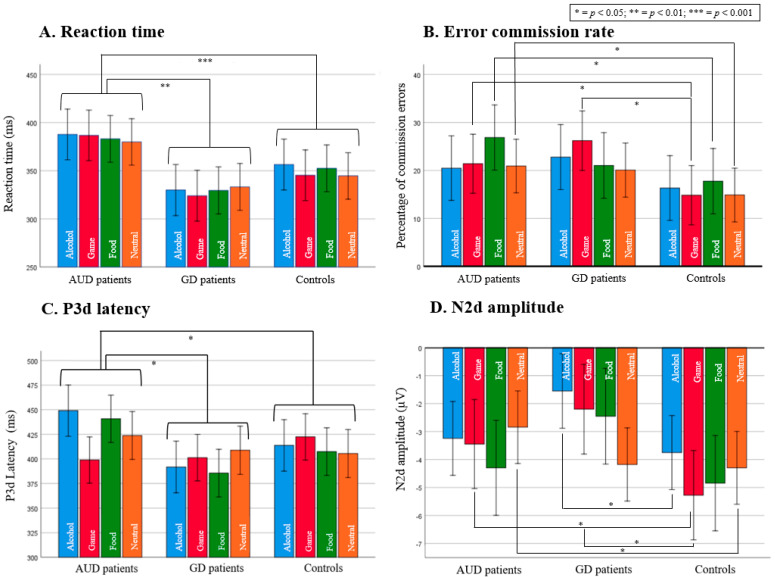
(**A**) Reaction time for correct detections of contextual Go/NoGo tasks; (**B**) Percentage of commission errors for each contextual Go/NoGo task; (**C**) P3d latency for each contextual Go/NoGo task; (**D**) N2d amplitude for each contextual Go/NoGo task. Error bars represent two standard deviations. Abbreviations: AUD, alcohol use disorder; GD, gambling disorder; ms, milliseconds; µV, micro volt.

**Figure 4 biology-12-00643-f004:**
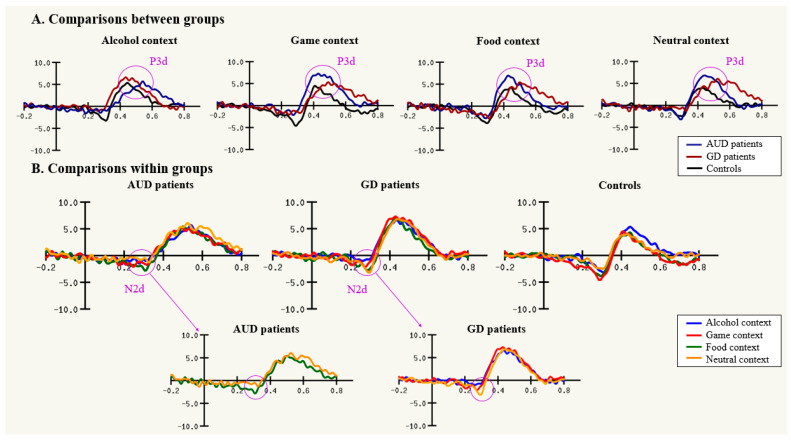
(**A**) Illustration of the event-related potential components of the inhibition responses (N2d and P3d) on the Cz electrode during contextual Go/NoGo tasks. (**B**) Comparison within groups between contexts with main differences on the N2d. The unit is in milliseconds on the ordinate and in microvolt on the abscissa. Abbreviations: AUD, alcohol use disorder; GD, gambling disorder.

**Table 1 biology-12-00643-t001:** Demographic and clinical characteristics. Abbreviations: AUD, alcohol use disorder; AUDIT, Alcohol Use Disorder Identification; BDI-II, Beck Depression Inventory II; CEQ, Craving Experience Questionnaire; GD, gambling disorder; M, mean; SD, standard deviation; SOGS, South Oaks Gambling Screen.

	AUD Patients *n* = 23		GD Patients *n* = 19		Controls *n* = 23		*F*/χ^2^	*p*	Post-Hoc Analyses
	M	SD	M	SD	M	SD			
Age	47.52	8.37	39.68	8.79	32.45	8.46	17.582	<0.001	AUD > GD > C
Education (number of years)	13.74	3.08	12.11	2.45	14.32	2.36	3.724	0.030	C > GD
Female/Male	8/15		1/18		9/13		7.195	0.027	AUD > G, C > GD
BDI-II	18.35	10.36	19.58	11.14	6.36	6.70	12.702	<0.001	AUD > C, GD > C
AUDIT	31.65	5.58	6.32	5.50	4.68	3.87	200.522	<0.001	AUD > GD, AUD > C
SOGS	0.26	0.62	12.42	3.47	0.09	0.29	258.244	<0.001	GD > AUD, GD > C
Alcohol CEQ									
intensity	24.39	12.59	/	/	/	/	/	/	/
frequency	24.57	13.23	/	/	/	/	/	/	/
Gambling CEQ									
intensity	/	/	40.80	17.70	/	/	/	/	/
frequency	/	/	35.53	14.64	/	/	/	/	/
Number of years of dependence	11.93	10.08	14.68	14.45	/	/	0.525	0.473	/
Number of previous alcohol detoxification	1.17	1.34	0.00	0.00	0.00	0.00	1.153	<0.001	AUD > GD = C

## Data Availability

The data that support the findings of this study are available upon request to the corresponding author.

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
