# Peer review of "A Comparative Event-Related Potentials Study between Alcohol Use Disorder, Gambling Disorder and Healthy Control Subjects through a Contextual Go/NoGo Task"

_biology, 2023, doi:10.3390/biology12050643_

Round 1

Reviewer 1 Report

The main aim of the study was, to compare the neurophysiological correlates of inhibitory processes when AUD and GD patients are performing a Go/Nogo task including a long-lasting context triggering a neutral, a natural rewarding (food), an alcohol-related or a gambling-related cued context.

The background literature review adequately justified the aim of the research. The findings presented in this paper add interesting data to the international literature on the topic, the paper itself should be of interest to a considerable interdisciplinary audience, including research scientists, practising therapists, psychologists and psychiatrists. The paper is well-written and have transparent composed. The statistical analysis used has been clearly stated. The results are well organized and presented.

The reading of the manuscript raises some concerns listed below:

- It is worth considering including a short paragraph presenting the limitations of the study and the application potential of the study results.

- Legends under tables are missing

After revision, including the aspects mentioned above, the article is fully eligible for publication in Biology Journal.  

Reviewer 2 Report

To measure the inhibitory mechanism implicated during the presence of related stimuli to their addiction, Dubuson et al are looking at ERP marker visible during a Go/NoGo task. The groups are patient suffering from Alcohol use and Gambling disorder, and they appear to both show poorer inhibitory performance, but some differences appear between the two groups regarding the context of the related stimuli. The manuscript is well written, the research question is well defined, and this is in line with the introduction. The results are well presented as the discussion except for 3.4 and a part of the discussion that are very difficult to read.

The introduction is well written. Citations from different research group and fields. Very easy to follow.

2nd intro paragraph, RI is it for Response Inhibition? So it could be stated on the second line of the intro.

p.3 line 12 ‘commission errors’ could be defined in a few words to help the reading.

Method

2.4 some more details about the EEG system like brand, model, companies would be needed.

2.5 typo IMB should be IBM?

2.5 can be a single paragraph instead of several small ones.

2.5 Mixed Anova is a good analysis, but the use of T-test as post-hoc appears questionable since SPSS could have been giving the post-hoc with proper controlling the Boneferroni error (or other bais). But the paragraph under the table seems to suggest that SPSS post-hoc has been used. So, it is probably ok but the way it is stated is not always clear and lead to think that T-Test were calculated after the Anova and not as a post-hoc controled for Boneferroni.

Results

The T test ddl should be noted like this t43 = 3.22, p = 0.002

F is supposed to be italicized

Thanks to have reported the np2.

Figure 3, to simplify the reading of the figure instead of marking p values with the exact numbers may be using stars to indicate * = p < 0.05, ** = p < 0.01, *** = p < 0.001

Fig 3D, need some works with the signification lines, they appear to touch the bars? The fig 3B is more like it should be.

Fig 3A and C, the Y axis is not starting at zero, this is a bit misleading, at least a clear broken Y axis should indicate it.

I do not see the point of having the Table 2, all the numbers in it are the numbers used to build the fig 3. The main effects of Context, Group or their interaction could be state in the text. Anyways, why the N2d and P3d are not in this Table. And why correct detection is in the table but not shown in the figure. In the first place, I do not see the value of this table.

In section 3.3, line 21:  are these post-hoc tests from the mixed Anova or some T-test made one by one? This comment goes with the previous one on the Boneferroni error.

Line 70: I think the N should be n

Section 3.4 Correlations. This section is very dense to read. Is there another way to present those results with a correlation matrix?

Discussion:

Line 94: not sure to follow the slower P3d latency in AUD and GD, fig 3C is not showing slower for the GD?

Section from lines 93 to 124: this part of the discussion is very dense and sound like results section. I do not have a solution for this, but I suggest rewriting this section to help the reader.

Reviewer 3 Report

The study investigates differences in inhibitory and rewarding processes between patients with alcohol use disorder (AUD) and gambling disorder (GD). The results suggest that AUD patients show a preserved inhibitory performance in the alcohol-related context, while GD patients show a specific inhibitory deficit in the game-related context. These findings have implications for therapeutic interventions for AUD and GD patients.

Study recruited 23 AUD inpatients, 19 GD patients and 22 healthy controls who underwent 4 separate Go/Nogo tasks, each incl.an alcohol, gambling, food and a neutral long-lasting cueing context while event-related potentials were recorded.

The paper provides a clear background of the study and highlights its importance in investigating potential differences in inhibitory and rewarding processes between AUD/GD pts. The methods sections is adequately described.

The soundness of the language is technical, scientific questions and hypothesis are stressed out in an acceptable way. The authors provide many references to support their claims in the text. Conclusions taken from the study are supported by the results. The presentation of the results is clear and understandable. The findings seem original and novel.

However, several points should be discussed;

1) Used references somewhat seem to be quite older with a detached view. Explain please that in the text or use more recent references.

2) The text often uses a number of misleading conjunctions that bring the reader back to the beginning of the sentence or paragraph. Just as an example (results in the abstract), where the first two sentences start the same way "On the other hand". Please revise the text and make it more readable for the readers.

3) In the course of the text we encounter a long-range analysis of repetitive statements and wordings. The introduction appears to be unnecessarily extensive, and at the same time the essential questions are opened only briefly. In the conclusions section, the results are often repeated without sufficient comparison with the sources, and the real impact on clinical practice is not adequately discussed. Moreover, the necessary critical re-evaluation of the entire study, including the methodology, is neglected. Is there a reason why individual limitations of the study are not thoroughly discussed (sample size, again methodology, generalization, etc...)?

4) Result sections is overwhelmed and oversaturated with data that doesn't necessarily show anything concrete. Could the individual parts be adjusted, reduced or presented more clearly?

Round 2

Reviewer 3 Report

The authors have responded adequately to the points raised and I firmly believe that all the modifications are in line with all the issues highlighted. The article can be accepted in this form.